# Agent-Based Modeling and Simulation of Tourism Market Recovery Strategy after COVID-19 in Yunnan, China

**Yumei Luo [1], Yuwei Li [1], Guiping Wang [1] and Qiongwei Ye [2,*]**

[1] School of Business and Tourism Management, Yunnan University, Kunming 650500, China; luoyumei@ynu.edu.cn (Y.L.); liyuwei236@163.com (Y.L.); Carolwgp@163.com (G.W.)
[2] Business School, Yunnan University of Finance and Economics, Kunming 650500, China
* Correspondence: yqw@ynufe.edu.cn

**Abstract:** The tourism industry hit severely by COVID-19 faces the challenge of developing effective market recovery strategies. Nonetheless, the existing literature is still limited regarding the dynamic evolution process and management practice. Hence, this study chose several famous spots in the Yunnan Province of China as the focus for a case study and utilized an agent-based simulation method for the decision-making process of tourists' destination selection and the dynamic recovery process of the destinations under different price and information strategies. The study found that the recovery effects of information strategies are positive, negative, or have no effect in different destinations. In contrast, price strategies can significantly stimulate an increase in the market share of destinations. When price strategy and information strategy are applied simultaneously, the interaction effects are inconsistent in different destinations. The findings contribute to the prediction of the recovery effect of strategies, can reduce trial and error costs, and can improve the scientific understanding of tourism market recovery.

**Keywords:** market recovery; agent-based modeling and simulation; destination decision process; recovery strategy

## 1. Introduction

With the rapid development of the economy and the gradual increase in consumption, tourism has become an important industry in the national economy [1]. However, tourism is an industry characterized by the spatial mobility of people. It is highly dependent on tourism demand, which is affected by economic, social and environmental changes, showing a high degree of sensitivity and uncertainty [1], making it extremely vulnerable to various crisis events [2]. Since the outbreak of COVID-19 in December 2019, national and international tourism activities around the world have been suspended due to the risk of infection by the epidemic, making 2020 arguably the most difficult year for the global tourism industry [3]. Arrivals in March 2020 dropped sharply by 57% following the start of lockdown in many countries, as well as the widespread introduction of travel restrictions and the closure of airports and national borders. This translates into a loss of 67 million international arrivals and about USD80 billion in receipts [4]. The World Bank estimated that the decline in global GDP would reach 3.9% under amplified global pandemic scenarios in 2020 [5]. The prognosis for the tourism sector alone is that there will be a decrease in output of 50–70% [6,7].

Under the impact of COVID-19, to prevent the spread of the epidemic, the Chinese government implemented different risk-control strategies (i.e., restrictions on mobility and travel, suspension of tourist services, postponement of the resumption of work). During the Spring Festival in 2020, the amount of revenue loss for enterprises in scenic spots in China was more than 90% of the level in the same period in 2019. The annual income loss of enterprises in Chinese scenic spots is expected to reach 40% to 50% of last year's income level [8]. From January to February 2020, hotels and B&B accommodation companies lost

more than 67 billion yuan in turnover, making them one of the most severely damaged industries [9]. With the epidemic under effective control in China, recovery of tourism has become a top priority during this period. As of the end of March 2020, according to data released by the Ministry of Culture and Tourism, the resumption rate of China's A-level scenic spots has exceeded 30%, more than 3700 A-level scenic spots have been opened, and the pace of recovery of the domestic tourism market is accelerating [5]. Therefore, how to develop effective tourism industry recovery strategies has become an urgent issue.

Research on tourism recovery strategies after crises or disasters has been fruitful. Among these studies, the most common concern disasters such as earthquakes [10,11], tsunamis [12,13], influenza [14], etc. In research on the recovery from crises, the methods adopted mainly include qualitative analysis, statistical analysis, case analysis, and literature review [15]. Proposed strategies mainly focus on the physical restoration, communication, and marketing of the destination. The restoration of destination attractions or infrastructure is the first priority [16], because it directly affects tourists' confidence and risk perception [14]. After disasters or crises, information disclosure [17] and media communication [15] are effective strategies because they can convey positive information to society [18]. With respect to marketing strategies, it has been shown that discounts, price subsidies [19], brand strategy [20], and celebrity endorsement [21] are beneficial to restore a destination's image, reputation, and visitor volume [15].

Existing literature can help tourism managers formulate recovery strategies, but there are still limitations regarding the dynamic evolution process and management practice [2]. First, tourism activities are a complex system with multiple elements which are interdependent and interacting [15]. However, most existing research is based on a static and single perspective, and ignores the dynamic change of influencing factors. Second, the existing studies on recovery strategies are mostly based on questionnaire surveys and experiments, and the evaluation and testing of the effectiveness of the strategies are insufficient. Third, insufficient consideration is given to the heterogeneity of tourists. Existing studies usually regard tourists as a whole or divide them into a small number of market segments. However, in tourism, tourists' economic conditions, tourists' preferences, their risk perception, and other factors will affect their behavior, and influence the effective implementation of tourism market management strategies [2].

Simulation approaches have been recognized as an effective tool for tourism management [22,23]. In particular, the agent-based model (ABM), a method for studying the interaction between agents within complex systems and between agents and the environment under certain rules in the space-time dimension [24], has been widely applied in tourism simulation research [12]. Unlike other analysis models, the ABM can better describe the tourist-oriented, the nonlinear and concurrent scenic management system, and can more fully consider individual heterogeneity and adaptability in behavior, show the interaction of agents, and assess the effects of different recovery strategies [2]. Therefore, application of the ABM provides an appropriate method for rehearsing tourism recovery strategies.

However, considering the recent COVID-19 situation, the literature on the agent-based tourist behavior simulation system is relatively scarce. Therefore, the main objective of this study comprises two elements: first, using the agent-based modeling method to build a tourist destination selection model, fully reflecting the heterogeneity of tourists and the dynamic development of the system, and considering in depth tourists' response to different recovery strategies; second, the model simulates the dynamic evolution process of the market share of the tourist areas after the implementation of different strategies, which assists scenic area managers to predict the effect of the strategies, reduce trial and error costs, find scientific and effective recovery strategies, and promote the faster recovery of the tourist market.

## 2. Literature Review

### 2.1. Tourism Crisis Management and Recovery Strategy

Tourism crisis management mainly focuses on the impact of crisis events caused by terrorism, war, political instability, crime, natural disasters, etc., on the tourism industry [25].In order to effectively manage the crisis, some scholars have developed a forward-looking and predictive framework for tourism crisis management based on the life-cycle of the crisis. The framework developed by Faulkner is widely used [26]. Faulkner divides tourism crisis management into six stages, including a pre-event stage, a predromal stage, an emergency response stage, a short-term recovery stage, a long-term recovery stage, and a resolution stage to achieve a new stable state. The framework has recently been used to evaluate infectious diseases [27], the recovery of tourism after COVID-19 [28], and tourist 'rip-off' incidents [2]. This study primarily considers the emergency response stage and the short-term recovery stage in the post-disaster reconstruction of tourist destinations. On the one hand, all kinds of crisis information are rapidly disseminated at these points, while crisis events continue to ferment [25]; if measures are not taken in time, the public's stereotyped negative image of a tourist destination can bring huge psychological repair costs. On the other hand, huge social pressure and the high cost of trial and error can cause managers to easily fall into a decision-making dilemma [29]; if intervention to address the crisis is not timely and effective, or the intervention strategy is incorrect, the negative impact of the crisis can be aggravated.

In order to effectively deal with the tourism crisis and restore the market, some scholars have suggested relevant strategies and suggestions. Gu et al. [22] adopted a system dynamics method to test the recovery effects of social distancing, tax reduction, travel bubbles, and joint strategies on the tourism market in the post-COVID-19 context. The results showed that the most effective strategy to change tourist behaviour intentions was the travel bubble strategy. Ming and Zhao [30] emphasized that brand remodeling is an important part of tourism crisis management, which is directly related to the recovery of consumer confidence after the disaster [20]. At the same time, formulating certain preferential reduction and exemption policies can promote the resurgence of people flow. Zhang et al. [31] combined econometrics and judgmental methods to predict the possible paths of Hong Kong's tourism recovery. They pointed out that tourism practitioners need to consider tourism reform and innovation, promote smart and digital tourism, and rebuild tourists' confidence in the tourism industry.

In addition, some research related to tourism crisis management has emphasized the importance of information strategy. Wu and Zhang [32] pointed out that rampant rumors and media dissemination of information about the epidemic have intensified people's fears, changed tourists' psychological expectations and behavior patterns, and directly led to drastic fluctuations in demand in the tourism market. Lehto et al. [18] found that safety information is the primary type of information communicated for tourism purposes. The dissemination of safety information can effectively address the stigma attached to tourist destinations and affect tourists' risk perception, and further influence tourists' destination selection [16], which is key to tourism recovery [15,33]. In response to the avian flu incident in Kyoto, Japan, Okuyama also found that information strategy was an important strategy for recovery [14]. Price strategy was also one of the commonly used recovery strategies. Through a literature review, Li [34] found that the benefits of price discounts in scenic spots can effectively offset tourists' perception of their own tourism risks, and the price promotion of well-known scenic spots immediately creates higher attractiveness. Gu et al. [22] found that the price strategy of tourism departments, by reducing tax on goods and services, can effectively increase the number of tourists. Laarman and Gregersen [35] also stated that the government, through price control, subsidies, etc., can confine the overall travel expenses of tourists within a certain range, which can help attract more tourists quickly in the short-term. It is apparent that both information strategy and price strategy play important roles in tourism crisis management.

*2.2. Agent-Based Model (ABM)*

The agent-based model (ABM) as applied to tourism research mainly focuses on four aspects: tourist flow management in scenic spots, tourism spatial layout, tourism planning, and tourists' destination decision-making. In terms of tourist flow management in scenic spots, Li [36] built an agent-based tourist behavior model and simulation environment for scenic spots by observing the passenger flow in the Summer Palace scenic spot during holidays, forecasted the temporal and spatial distribution of tourist flow in scenic spots, and carried out analysis of the dynamic control of passenger flow. In terms of the layout of tourism space, Yuan and Zheng [37] developed a model to predict the spatial and temporal distribution of tourists by using a Markov model to alleviate crowding in theme parks and constructed an agent-based model for verification. The results showed that the predictive effect of the model was better than existing methods. In tourism planning, the agent-based method has also received extensive attention. Johnson and Sieber [38] built a tourist-destination ABM model for Nova Scotia, Canada, and explored the dynamic changes of tourist volume in three scenarios: keeping the status quo, experiencing an economic crisis, and using advertising strategies. Balbi et al. [39] established an ABM model of tourists and destinations for the tourism development plan of a European ski resort during change of climate and tourism demand. Considering eight types of tourist subjects, the corresponding parameters and behavior rules were set and simulated to verify the effectiveness of the different tourism development strategies. In terms of tourist destination decision-making, Boavida et al. [40] constructed a tourist destination decision-making model based on the ABM. Considering the tourist's motivation, rational behavior rules, emotion and satisfaction, as well as the influence of social networks, the model simulated the decision-making process of tourists' choice of the Alentejo coastal resort as a destination. Alvarez and Brida [24] also constructed a dynamic ABM model for the evolution of tourist destination decision-making, which simulated the dynamic evolution process of tourist destination decision-making based on variables such as tourists' preference for products and services, crowding types, and individual inertia levels.

In sum, it has been shown that the ABM is very suitable for the study of tourism systems and has unique advantages in revealing the decision-making process of tourist destination choice and the dynamic evolution of the tourism system. However, the research on tourism crisis management using ABM remains very limited. Most existing tourism crisis management research is based on a static and single perspective and ignores the dynamic interaction process of the internal elements of the system. In addition, existing studies mainly apply the ABM method to tourism research under normal conditions, while there is a lack of research on the application of ABM method to the situation of COVID-19. Therefore, this study adopts ABM to explore the decision-making process underpinning tourists' destination selection following COVID-19 from a dynamic perspective, and illustrates the dynamic recovery process of the destination for different types of recovery strategy, such as information and price strategies.

## 3. Methodology

*3.1. Case Description*

Tourism is one of the largest industries in the world today, and it is also a sensitive industry that is particularly vulnerable to the external environment. As one of the important tourist provinces in China, Yunnan has a good tourism image and is one of the most popular tourist destinations. The province has more than 200 scenic spots, of which 134 are national A-level and above and 9 are 5A-level scenic spots, with rich tourism resources. After decades of development, tourism has become one of the industry pillars of Yunnan Province's economy [24]. According to the statistics of Yunnan Province in 2019, there were 814.06 million tourist trips in Yunnan, and tourism revenue reached approximately CNY 1.10 trillion. However, in January 2020, during the Spring Festival travel season, COVID-19 suddenly appeared. All tourist attractions and services in Yunnan Province were suspended from 24 January. The number of tourists in the tourist attractions dropped

precipitously. In Kunming, Yunnan Province, for example, during the Spring Festival Golden Week in 2020, the city received a total of 1.84 million domestic and overseas tourists, down by 81.84% year-on-year; the total tourism revenue of the city was CNY 1.512 billion, down by 76.52% year-on-year. As one of the industry pillars, tourism plays an important role in Yunnan's economy, and the negative impact of COVID-19 on tourism will inevitably affect the economic development of Yunnan. Therefore, this study focuses on five tourist destinations with 5A-level scenic spots (i.e., Lijiang, Dali, Baoshan, Xishuangbanna, Diqing), as the case study objects, and constructs a tourism competition market simulation system to explore the tourism recovery strategy of Yunnan Province after COVID-19. These destinations with high popularity, reputation, visit and re-visit rate, and belonging to the brand scenic spots of Yunnan, are good representatives for the development of Yunnan tourism. The geographical location of Yunnan and the five tourist destinations (Lijiang, Dali, Baoshan, Xishuangbanna and Diqing) are shown in Figures 1 and 2.

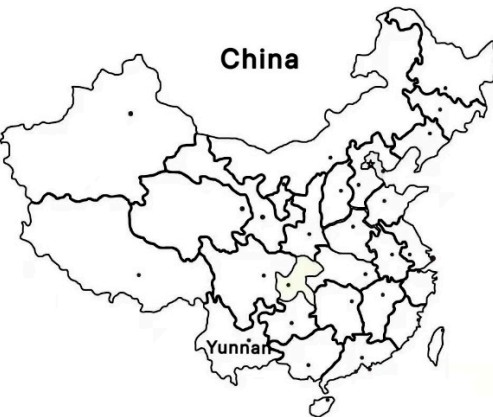

**Figure 1.** The location of Yunnan (source https://quark.sm.cn/api/rest?method=Picbig.newHome&format=html&schema=v2&query=%E4%B8%AD%E5%9B%BD%E5%90%84%E7%9C%81%E9%9D%A2%E7%A7%AF%E5%9C%B0%E5%9B%BE&uc_param_str=dnntnwvepffrgibijbprsvdsdicheiniut&hid=nlMKqkVLU9CAifgiiA5RWYnr6ICA6qvo&from=share&share=1&pk=http://www.51wendang.com/pic/77da5ca74bb14b411d28e6ec/1-810-jpg_6-1080-0-0-1080.jpg (accessed on 21 October 2021)).

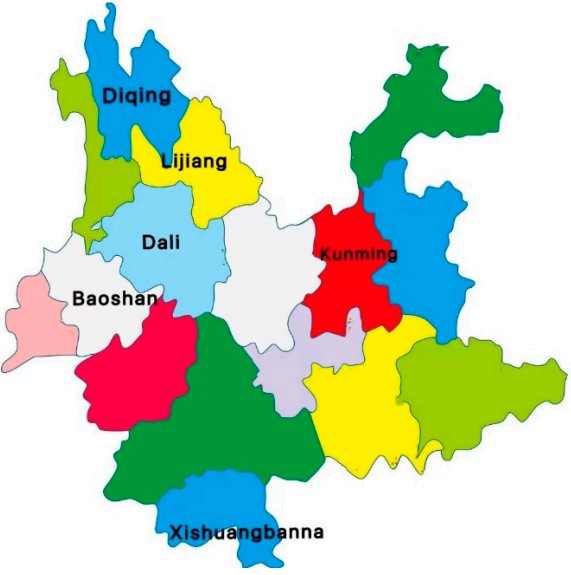

**Figure 2.** The location of five tourist destinations (Lijiang, Dali, Baoshan, Xishuangbanna, Diqing). (source: http://www.ljshuhe.com/view/ljdypc/1/92/view/184.html (accessed on 21 October 2021)).

### 3.2. Developing an Agent-Based Model

This paper utilizes an agent-based simulation method and NetLogo logging 6.0.4 to comprehensively consider the internal and external influencing factors of destination decision-making; the agent-based model is carried out to simulate the destination decision-making process of tourists. Using the model, this paper investigates the recovery of regional market share after the epidemic under different levels of information strategy, price strategy, and their combination. The strength of information strategy and price strategy were separately divided into three groups: none, low, and high.

The research on the decision-making model of tourism destination primarily incorporates: (1) a tourist decision-making process model, (2) a choice sets model, (3) a theoretical model of planned behavior, (4) a discrete selection model, and (5) a multi-factor influence model based on stimulus-organism-response (SOR) theory. Among these, the SOR multi-factor influence model can comprehensively consider a variety of factors that affect tourists' destinations selection, including macro- or micro-level factors that influence the choice of tourist destinations [41]. Hence, this study adopts a multi-factor influence model based on SOR theory, dividing the factors influencing destination decisions after COVID-19 into tourists' internal factors and external factors, as shown in Figure 3.

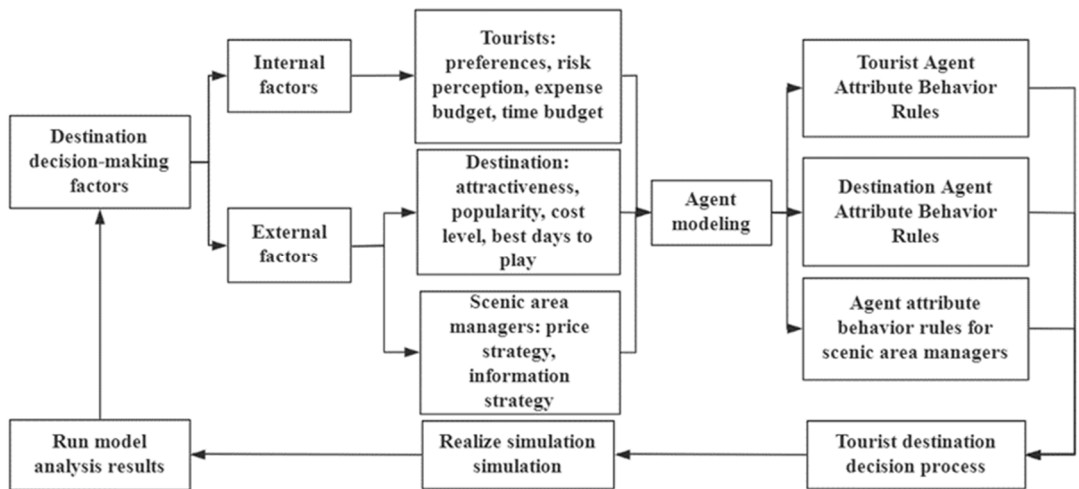

**Figure 3.** Framework of tourism destination decision-making model based on the agent.

As shown in Figure 3, the internal factors affecting tourists' destination selection include tourists' preference, risk perception, time budget, and expense budget. The external factors include destination factors and scenic-manager-related factors. Among these, the destination-related factors include attractiveness, popularity, cost level, and the best days to visit, while factors related to scenic managers include price strategy and information strategy.

A comprehensive consideration of the internal and external influencing factors of destination decision-making, applying the agent-based model was carried out for tourists, destinations, and scenic managers. Specifically, tourists evaluate the attraction, risk and cost of candidate destinations according to their own preference, risk perception, time budget and expense budget, combined with attractiveness, popularity, cost level, and the best visiting days of each destination, then make a choice. In the context of the epidemic, the information strategy of scenic managers is taken to act on tourists' risk perception, and the price strategy to act on the cost level of the destination. The specific principles of the model design are described below.

### 3.2.1. Tourist Agent

Tourism destination selection is a decision-making process influenced by many factors, including tourists' internal factors that affect their cognitive and emotional judgments,

as well as external factors relating to tourism destinations [41].These two factors jointly affect the whole decision-making process of tourist destination selection. Therefore, the tourist agent must be analyzed from two points of view: tourist attributes and destination attributes. Specifically, the main factors affecting tourists' destination selection include tourists' preferences, budget, time constraints, tourism prices, tourism product quality, and risk perception [42,43]. Therefore, for the tourist agent, this study considers the following four main attributes: (1) preference, (2) risk perception, (3) expense budget, and (4) time budget. Preference refers to the degree to which tourists are interested in and willing to visit a tourism destination, which may be manifested as the frequency of choosing a particular tourism destination [2]. And the more comments on the scenic spot, the more people are interested in the scenic spot. Potential tourists can also obtain a more comprehensive understanding of the situation of the scenic spot from the numerous comments which promotes its choice. Risk perception refers to the tourists' comprehensive cognition of various risks that may occur in tourist destination [2] and has a significant impact on the tourists' selection of a destination [44]. Expense budget refers to the amount of money a tourist is willing to spend at a certain destination. Time budget refers to the maximum number of days a tourist is willing to stay in a tourist destination. According to the law of large numbers and the central limit theorem, when the number of observed objects is large (the number of tourists is n > 1,000,000), the numerical distribution of these attributes can be approximated to a normal distribution [2]. Therefore, this study assumes that the numerical distribution of the four attributes of tourists all follow a normal distribution to fully reflect the heterogeneity among tourists.

### 3.2.2. Destination Agent

Each destination contains different types of scenic spots. According to Geldner's scenic spot classification theory, this study divides the scenic spots into five types, including culture, nature, festivals, recreation, and entertainment. Each destination fits one or more of the above five categories. Corresponding to the properties of the tourist agent, this study also considers four attributes of the destination, including attractiveness, popularity, cost level, and best days to visit. Attractiveness refers to the extent to which the destination can satisfy tourist's interests [45]. The more a destination meets the interests of tourists, the more attractive it will be to tourists and the more likely it will be selected as a potential destination. Popularity refers to the degree to which the destination has been heard of by tourists. Popularity is a primary prerequisite for tourists to select the destination. The higher the popularity of the destination, the more likely tourists will select it [43]. Cost level is defined as the historical average daily cost to tourists in the specified destination. The best days are the optimal days for tourists to visit a certain destination.

### 3.2.3. Tourism Market Recovery Strategy

Price strategy is one of the recovery strategies commonly used in public health management, which plays an important role in the recovery of tourism markets. In China, the government's price strategy most often includes lowering or discounting the ticket price of scenic spots, strengthening the supervision of the market price in scenic spots, providing special subsidies, etc. [46]. Reducing the cost of tourism can attract more tourists and encourage them to spend extra money in other parts of scenic spots [34]. When the ticket price is kept at a relatively low level, tourists will feel "excellent value for money" [15], and such benefits perceived by tourists can effectively offset the perception of tourism risks and affect their willingness to travel [34]. Price reduction of scenic spots enables consumers to form a vertical comparison, which improves their purchase intention [34]. Therefore, with respect to the discount level of daily per capita cost level of the destination, two kinds of discounts (i.e., low and high) were set for the price strategy in the model to study the market recovery effect of price strategy with different discounts.

Positive information strategies play an important role in eliminating the negative impact of the tourism crisis and reducing tourists' perception of risk [15]. Safety information

can alleviate the anxiety of tourists and offset any negative publicity about the destination, thereby protecting the destination's image [15]. In the crisis context of earthquakes, SRAS, and overcharging of tourists, studies found that the government generally adopted information strategies to reduce the negative impacts [15]. However, Zheng [47] suggested that the process of crisis information dissemination includes two closely related subjects and behaviors. One is the responsible subject for resolving the crisis, namely, the government, and its corresponding behaviors, namely, information disclosure; another is the communication subject of crisis resolution, namely, the media, which corresponds to news reports behavior [47]. Therefore, the effectiveness of information strategy implementation is not only affected by government investment, but also by the media and information receivers. This model includes two variables, media attention and tourist information involvement, to study the effectiveness of government information strategy.

### 3.2.4. Decision-making Rules for Tourists' Behavior

Assume a tourist T, with cost budget Tc, time budget Td, and best days of visit of the *i*th scenic spot, Di. The specific rules are shown in Figure 4.

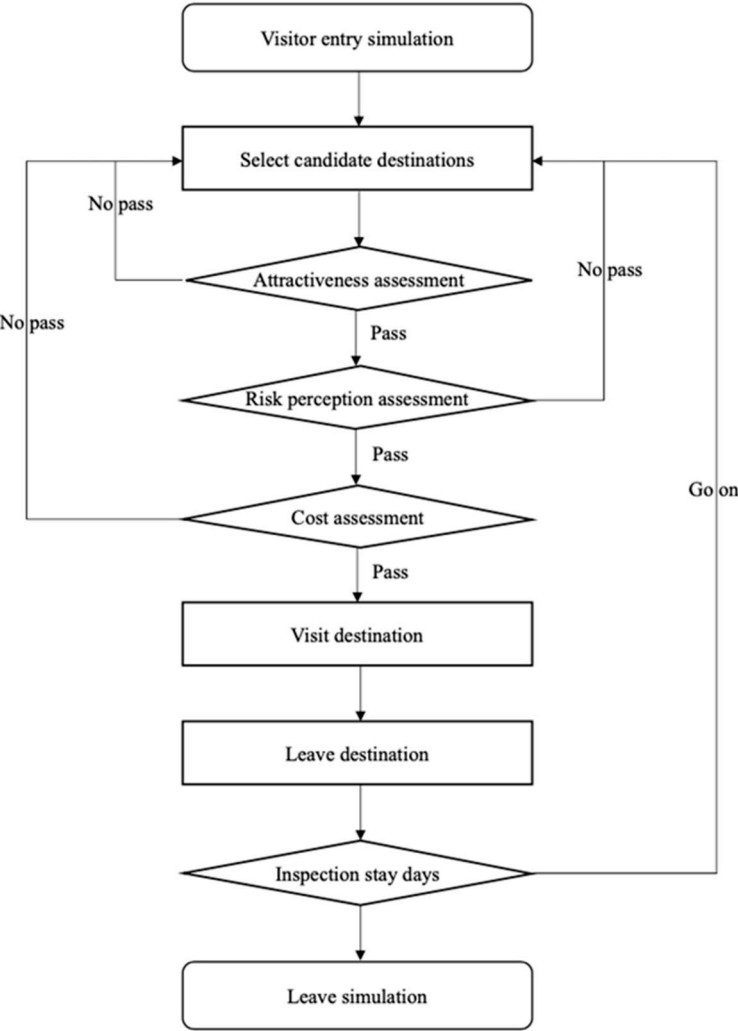

**Figure 4.** Decision-making rules for tourists' behavior.

- Select candidate destinations

There are six candidate destinations in the system, including Kunming, Lijiang, Dali, Baoshan, Xishuangbanna, and Diqing. The selected probability of each destination is set

according to its popularity which is measured by historical tourist reception. Tourist T selects a candidate destination P and carries out various evaluations on it.

- Attractiveness assessment

The system assumes that the attractiveness of each destination to tourists is determined by the quality of scenic spots in each region consistent with tourists' preferences. The probability that the candidate destination passes the tourists' attractiveness evaluation is $P_k = S_k/S_{max}$, where $S_a$ is the score of scenic spot K of destination P, and $S_{max}$ is the highest score of the scenic spot, that is, the highest score set by the rating system. If the assessment is passed, the tourist moves to the next assessment; otherwise, returns and chooses a new destination.

- Risk perception assessment

The system assumes that the risk perception of tourists follows the normal distribution of $N(\mu,\sigma)$ to reflect the heterogeneity among tourists. It is assumed that the risk perception threshold in the system is $R_t$ and the current risk perception of tourist T is $R_p$. When $R_p < R_t$, the risk perception of tourist is at an acceptable level and the next assessment is carried out; otherwise, returns and chooses a new destination.

- Cost assessment

The system assumes that the daily cost of candidate destination P is $c_P$ and the best days to visit are $d_P$. When $T_c - (c_P \times d_P) \geq 0$ and $T_p \geq d_P$ (where $T_c$ is expense budget of tourist and $T_p$ is the time budget of tourist), tourists can visit the candidate destination P; otherwise, returns and chooses a new destination.

- Visit destination

Update the actual dwell days $d_t$ with the change of simulation unit time. When the above assessments are passed, tourist T will visit the destination P. When $d_t = d_P$, leave the destination and update the tourist's expense budget $T_c' = T_c - (c_P \times d_P)$ and time budget $T_p' = T_p - d_P$. When the remaining time budget of tourists $T_c'$ is greater than or equal to the minimum optimal visiting days in the system, i.e., $T_p \geq \min\{d_1, d_2, \ldots \ldots, d_n\}$, tourists can select the candidate destination again.

### 3.3. Data Collection

Ctrip.com has long been in a leading position and has high popularity in the field of tourism e-commerce in China. Therefore, the data of this paper comes from www.ctrip.com and official statistics (see http://stats.yn.gov.cn/tjsj/tjnj/, accessed on 11 April 2021). In this paper, the data of scenic spot preference, expense budget, time budget, attractive force, best days to play, and daily fee are from ctrip.com, while the data on popularity is from the official government website. Table 1 shows the initial values, data sources, and attribute measurement of each parameter in the model.

**Table 1.** Initial values, data sources and calculation methods of main parameters.

| Agent | Attribute | | Initial Value | Data Sources | Measurement of Attributes |
|---|---|---|---|---|---|
| sTourists | Scenic spot preference (s) | Natural | 0.643 | Ctrip.com | $s = c1/c2 * 100\%$ c1 is the comments of all destination scenic spots under a certain category of activities. c2 is the number of comments under all categories of activities. |
| | | Entertainment | 0.028 | | |
| | | Culture | 0.263 | | |
| | | Recreation | 0.066 | | |
| | Expense budget | Mean value | 3174 | Ctrip.com | Per capita cost marked in travel notes issued by underground tourists at each destination |
| | | Standard deviation | 857 | | |
| | Time budget | Mean value | 5 | Ctrip.com | Travel days marked in travel notes issued by underground tourists at each destination |
| | | Standard deviation | 2 | | |
| Destination | Attractive force | Kunming | 4.55 | Ctrip.com | Average score of scenic spots under this activity type at each destination |
| | | Dali | 4.5 | | |
| | | Lijiang | 4.6 | | |
| | | Xishuangbanna | 4.47 | | |
| | | Diqing | 4.67 | | |
| | | Baoshan | 4.5 | | |
| | Popularity (p) | Kunming | 0.403 | Government official website | $p = r1/r2 * 100\%$ r1 is the historical tourist reception of each region. r2 is the total historical tourist reception of the six regions. |
| | | Dali | 0.168 | | |
| | | Lijiang | 0.167 | | |
| | | Xishuangbanna | 0.103 | | |
| | | Diqing | 0.088 | | |
| | | Baoshan | 0.071 | | |
| | Best days to play (d) | Kunming | 3 | Ctrip.com | |
| | | Dali | 3 | | |
| | | Lijiang | 3 | | Set the number of days according to the suggestions of Ctrip.com |
| | | Xishuangbanna | 3 | | |
| | | Diqing | 3 | | |
| | | Baoshan | 3 | | |

**Table 1.** *Cont.*

| Agent | Attribute | | Initial Value | Data Sources | Measurement of Attributes |
|---|---|---|---|---|---|
| | Daily fee (f) | Kunming | 284 | Ctrip.com | $f = t/d$ t is the travel expenses. Which included tickets, transportation, accommodation(a), catering(c) and cableway. The transportation fee is the high-speed railway fee from Kunming to various regions. $a = h * (d - 1)$, h is the average price of the hotel published by Ctrip.com online. $c = r *$ two meals $* d$, c is the average price of restaurants published by Ctrip.com online. |
| | | Dali | 392 | | |
| | | Lijiang | 405 | | |
| | | Xishuangbanna | 376 | | |
| | | Diqing | 444 | | |
| | | Baoshan | 429 | | |

## 4. Model Testing

### 4.1. Simulation Environment and Related Parameter Settings

This paper uses NetLogo6.0.4 software to simulate the process of tourism destinations selection by tourists to six destinations every year. The simulation model takes days as the time unit, 12 months as a cycle, and the initial tourists are generated at the beginning of each cycle. In 2014, the total number of tourists in the above six regions was 152.7791 million. Referring to this data, this system set up 10,000 initial tourist agents, and each tourist agent represents 15,278 tourists; according to the average growth rate of tourist volume in each region from 2014 to 2019, the system sets an average annual growth rate of tourists as 21.15% to simulate the natural growth of tourists in the tourism market.

### 4.2. Simulation System Verification

Firstly, 10 tourist agents were set up to test whether tourist behavior operates according to the expected rules. Then the model sensitivity was tested by adjusting the simulation parameters of key attributes in the system. After 100 independent experiments, the results showed that the program could meet the research needs.

As shown in Figures 5 and 6, and Table 2, the model compared the actual data with simulation data on tourist number and market share in the six regions from 2015 to 2019 to verify the validity and robustness of the model. In order to control the influence of the uncertainty of the simulation output, the simulation takes the average value of 20 runs of the system operation results.

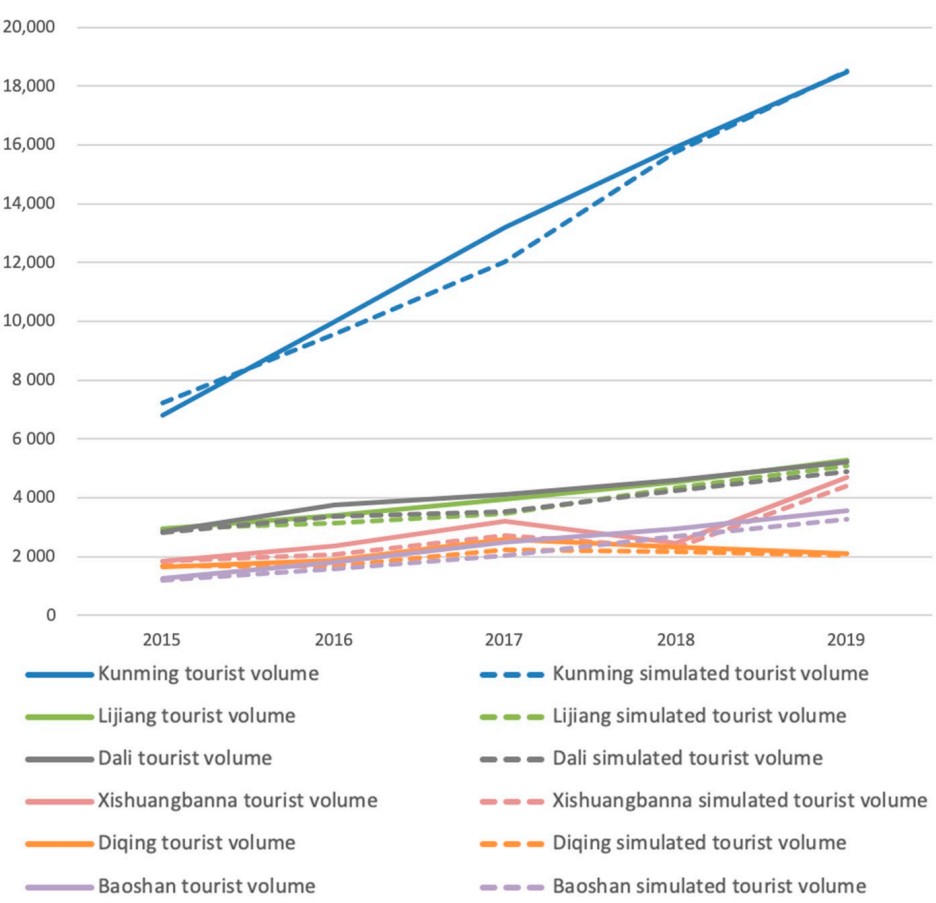

**Figure 5.** Simulation test diagram of tourist volume (10,000 people).

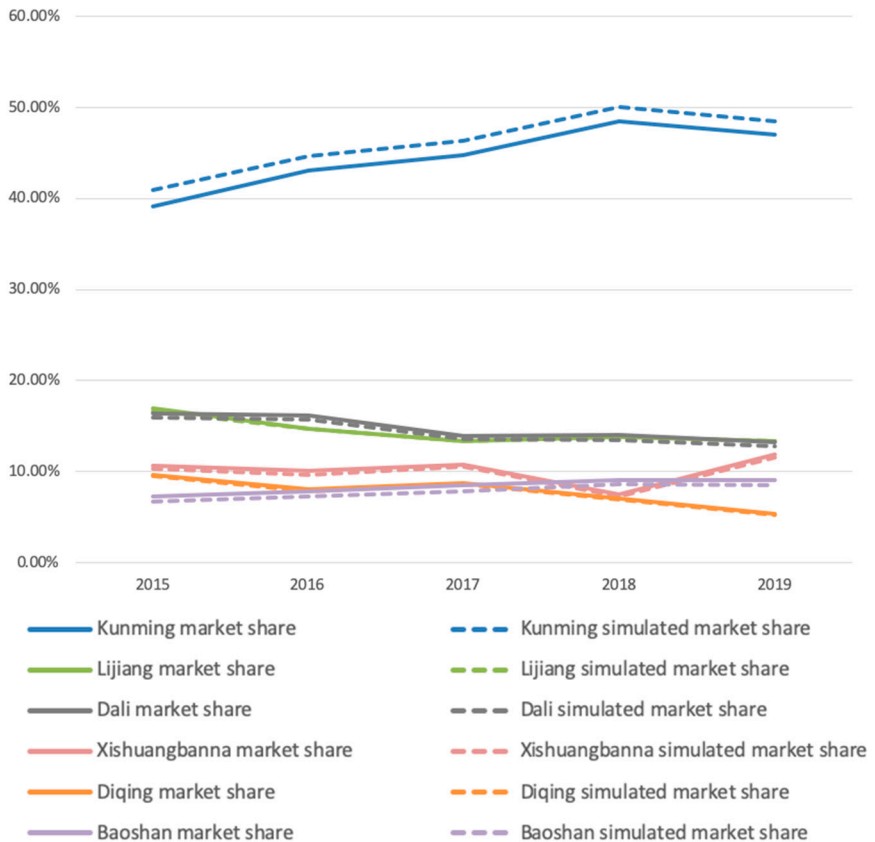

**Figure 6.** Market share simulation test chart.

**Table 2.** Comparison between actual data and simulated data of tourist volume and market share.

| Destination | Index | 2015 | 2016 | 2017 | 2018 | 2019 |
|---|---|---|---|---|---|---|
| Kunming | Tourist volume | 6796.91 | 9990.15 | 13,208.45 | 15,911.23 | 18,494.59 |
| | Simulated tourist volume | 7234.82 | 9568.08 | 12,031.43 | 15,752.08 | 18,500.82 |
| | Relative error | 6.1% | −4.4% | −9.8% | −1.0% | 0.0% |
| | Market share | 39.1% | 43.1% | 44.7% | 48.5% | 47.0% |
| | Simulated market share | 40.9% | 44.6% | 46.3% | 50.0% | 48.5% |
| | Relative error | 4.30% | 3.45% | 3.38% | 2.97% | 3.02% |
| Lijiang | Tourist volume | 2941.44 | 3404.1 | 3950.87 | 4523.88 | 5293.87 |
| | Simulated tourist volume | 2941.55 | 3152.92 | 3452.37 | 4335.82 | 5084.14 |
| | Relative error | 0.00% | −8.0% | −14.4% | −4.3% | −4.1% |
| | Market share | 16.9% | 14.7% | 13.4% | 13.8% | 13.4% |
| | Simulated market share | 16.6% | 14.7% | 13.3% | 13.8% | 13.3% |
| | Relative error | −1.65% | 0.07% | −1.02% | −0.22% | −0.60% |
| Dali | Tourist volume | 2841.26 | 3765.77 | 4120.45 | 4606.27 | 5206.51 |
| | Simulated tourist volume | 2813.14 | 3368.49 | 3522.11 | 4238.58 | 4892.70 |
| | Relative error | −1.00% | −11.8% | −17.0% | −8.7% | −6.4% |
| | Market share | 16.4% | 16.2% | 13.9% | 14.0% | 13.2% |
| | Simulated market share | 15.9% | 15.7% | 13.6% | 13.5% | 12.8% |
| | Relative error | −3.08% | −3.09% | −2.55% | −3.97% | −3.00% |
| Xishuangbanna | Tourist volume | 1859.53 | 2350.38 | 3205.07 | 2469.42 | 4704 |
| | Simulated tourist volume | 1831.14 | 2087.28 | 2734.38 | 2289.48 | 4405.41 |
| | Relative error | −1.55% | −12.6% | −17.2% | −7.9% | −6.8% |
| | Market share | 10.7% | 10.1% | 10.8% | 7.5% | 11.9% |
| | Simulated market share | 10.3% | 9.7% | 10.5% | 7.3% | 11.5% |
| | Relative error | −3.43% | −3.75% | −2.76% | −3.16% | −3.07% |

**Table 2.** *Cont.*

| Destination | Index | 2015 | 2016 | 2017 | 2018 | 2019 |
|---|---|---|---|---|---|---|
| Diqing | Tourist volume | 1658.18 | 1890.35 | 2582.89 | 2320.2 | 2110.08 |
| | Simulated tourist volume | 1677.37 | 1683.41 | 2230.21 | 2181.09 | 2024.11 |
| | Relative error | 1.14% | −12.3% | −15.8% | −6.4% | −4.2% |
| | Market share | 9.6% | 8.1% | 8.7% | 7.1% | 5.4% |
| | Simulated market share | 9.5% | 7.9% | 8.6% | 6.9% | 5.3% |
| | Relative error | −1.21% | −3.12% | −1.34% | −2.38% | −2.08% |
| Baoshan | Tourist volume | 1264.15 | 1816.44 | 2492.28 | 2960.77 | 3579.46 |
| | Simulated tourist volume | 1188.55 | 1571.95 | 2040.68 | 2704.21 | 3259.26 |
| | Relative error | −6.36% | −15.6% | −22.1% | −9.5% | −9.8% |
| | Market share | 7.3% | 7.8% | 8.5% | 9.1% | 9.1% |
| | Simulated market share | 6.7% | 7.3% | 7.8% | 8.6% | 8.5% |
| | Relative error | −8.63% | −6.27% | −8.42% | −6.12% | −6.62% |

Note: Relative error = (Simulation data–Empirical data)/Simulation data.

The results showed that relative error was controlled within a certain range; the maximum error was no more than 22.1%. The system reduces the initial number of tourists by 50% and simulates again to test whether the simulation results of destination market share are affected by the numerical changes of tourists. The two simulation results were basically consistent. Therefore, the following will focus on market share. Since the market share of Kunming differs greatly from the other five regions, only Kunming was used for verification and simulation environment setting, while the simulation results of the other five regions were compared and analyzed.

## 5. Scenario Testing: Analysis of Simulation Results

The epidemic reached a serious period of large-scale and rapid outbreak on 24 January 2020 [48,49] and all scenic spots in Yunnan Province were closed. On 20 February 2020, according to a guideline issued by the Yunnan Association of Tourist Attractions, all tourist attractions in Yunnan province were to be opened in an orderly manner after being submitted for approval. Therefore, the time of this simulation model starts from the first day after the opening of the scenic spot after the epidemic. The simulation environment and relevant parameter settings follow the initial value of model parameters, and the initial value of market share adopts data for 2019. Among these, information strategy acts on the risk perception parameters of tourists, and price strategy acts on the cost level parameters of destinations.

### 5.1. Information Strategy Analysis

Government and scenic managers sought to adopt a series of information strategies to alleviate tourists' risk perception affected by the outbreak of COVID-19, such as improving the popularity and attractiveness of the scenic spots and conveying information about the safety of scenic spots. During the epidemic prevention and control period, the managers of tourist attractions used online and offline channels to carry out online live broadcasting, online virtual scenic spots and smart tourism, and developed new tourism models, such as self-drive tourism, short-distance tourism, rural tourism, and theme series interactive experiences. At the same time, in order to realize epidemic prevention and control, all regions controlled the passenger flow and implemented ticket reservation management systems over time. All regions actively promoted information related to the epidemic to tourists, such as emergency rescue information and early warning of passenger flow and dispersal guidance information.

The average risk perception after the COVID-19 outbreak is revised by the intensity of information strategy and other factors, and is as follows: $\mu' = \mu + s \times w \times e^{-(p+0.05)t}$ where $\mu'$ is the average risk perception of tourists after the outbreak of the epidemic, $\mu$ is the average risk perception of tourists before the epidemic, s is the degree of media attention, w is the degree of information involvement of tourists, p is the strength of information strategies, and t is the number of days from the first opening day of the scenic spot after

the epidemic. Since Academician Zhong Nanshan pointed out that there was "human-to-human transmission" of the COVID-19 virus, media coverage of COVID-19 has entered a critical period. The number of confirmed cases, the number of deaths, the number of cured cases, and all kinds of information has been published [50]. During the outbreak of the epidemic, especially in the early stage of large-scale media coverage, potential tourists pay the most attention to the epidemic information [51]. Therefore, the simulation system set media attention and tourist information involvement at a stable and high level during the COVID-19 outbreak.

In this study, the strength of information strategy was divided into three groups: none, low, and high, and they were recorded as p0, p1, and p2, respectively. Figure 7 shows the market share recovery effect in different regions with different information strategy strength.

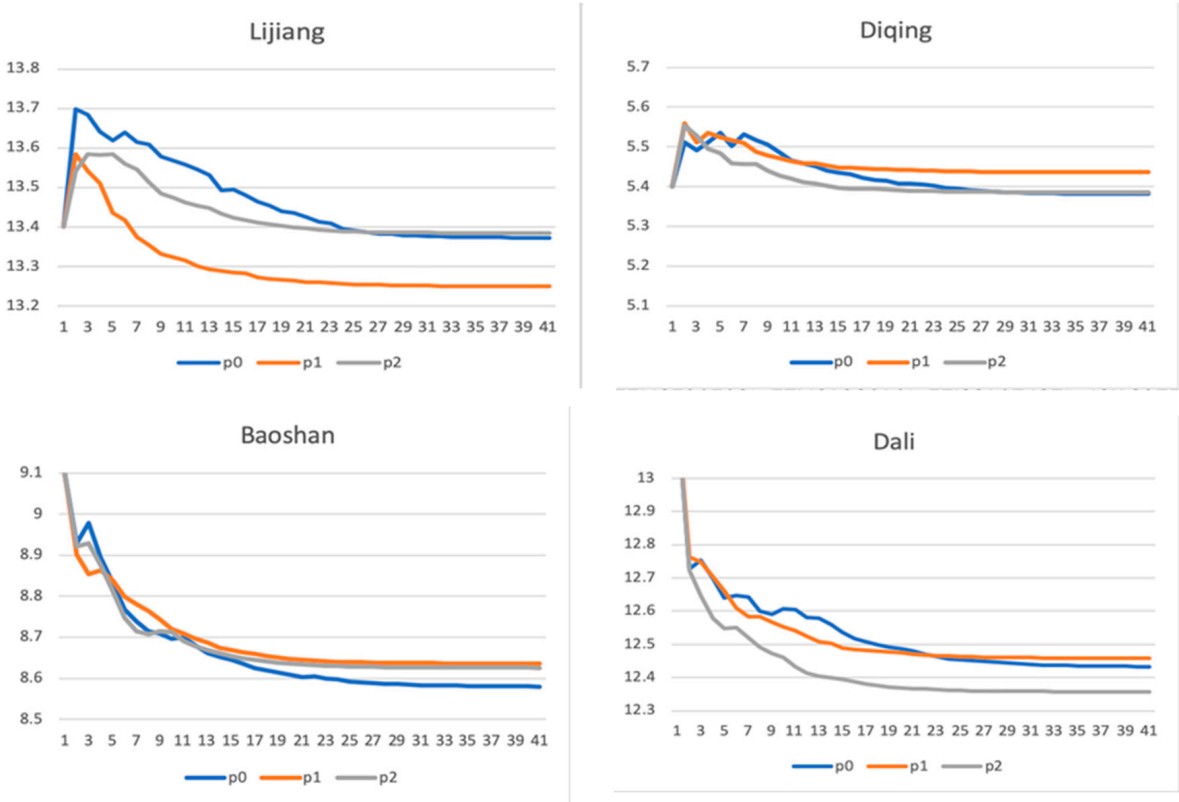

**Figure 7.** Market share of each region under different information strategies.

It can be seen from Figure 7 that the information strategy can restore the final stable market share of each region, but the recovery effect is small. Among the strategies, the p2 strategy is ideal in Lijiang, and its market share remains stable from 18 days after the opening of the scenic spot, while under the p0 strategy, the market will stabilize after 26 days, and the market share of p1 strategy will decline the most; in Diqing, the p1 strategy will increase its market share the most compared with the p2 strategy; in Baoshan, the recovery effect of the p1 and p2 strategies is basically the same; in Dali, the p3 strategy caused the biggest market share decline, while the effect of strategy p1 fluctuated up and down with p0, and exceeded p0 in about 23 days; in Xishuangbanna, the market recovery effect of p0 strategy is the best. In general, the results suggest that Xishuangbanna does not need to adopt an information strategy, Lijiang should adopt a p2 strategy, and other regions should adopt a p1 strategy. Therefore, the study found that the recovery effects of information strategies are positive, negative, or have no effect in different destinations.

After the epidemic, the market share of Lijiang and Diqing increased first and then decreased, and the market share of Dali, Xishuangbanna, and Diqing showed a downward trend. Irrespective of whether an information strategy is adopted or not, the markets in Lijiang and Diqing will improve. In the rising situation, the market share will rise first and then decline. In this process, Lijiang can adopt a no information strategy or a p2 strategy, while Diqing should adopt a p1 strategy. The possible explanation is that both regions are highly attractive and there is strong competitiveness in the market. Therefore, after the scenic spot is opened, whether an information strategy is adopted or not, the market share will be improved. With the passage of time, the epidemic is gradually brought under control, the risk perception level of tourists is gradually stabilized, and the market reverts to a normal level. Therefore, in this process, due to the high attractiveness and popularity of Lijiang, an information strategy need not be adopted, while Diqing has low popularity but high attractiveness, so only a low-intensity information strategy is needed to improve the market.

However, for other regions, information strategies play different roles in the situation of declining market share: first, the market share of each region is declining; second, for the strategies of p0 and p1 in Dali, the market share decreased less, while p2 decreased more; third, in Xishuangbanna, the market share decreased the least under the p0 strategy, while the stronger the information strategy, the more the market decreased; fourth, the market share of Baoshan with an information strategy is more than that without an information strategy, and the information strategy with different strength does not show different effects. The possible explanation is that the overall market is greatly affected by the epidemic because the attraction of the three regions is relatively low. Therefore, in this process, due to the low attraction and popularity of Baoshan, an information strategy needs to be adopted, while Dali and Xishuangbanna have relatively low attraction and high popularity, so it is recommended that they adopt a low-strength strategy or that they do not adopt one. Xishuangbanna is located at the border, and government intervention will expand users' risk perception, so it is more appropriate not to adopt an information strategy.

It can also be seen from Figure 7 that the different strength of information strategy has little impact on the market share of Baoshan and Diqing and has a greater impact on the other three regions. It may be because the popularity of Baoshan and Diqing is relatively low, and the impact of an information strategy on their market share is limited.

*5.2. Price Strategy Analysis*

This study sets three government price strategy scenarios of no, low, and high, corresponding to price discounting of 0%, 30%, and 70% of tickets, cableways, and accommodation in the tourism expenses in each region, respectively. Figure 8 shows the effect of market share changes when all regions adopt the price strategy.

It can be seen from Figure 8 that after the epidemic, the adoption of a price strategy has a positive effect on the recovery of market share in various regions, with the greater the strength of price strategy, the better the recovery effect of market share.

For Lijiang and Diqing where the market shares first increased and then decreased, the market share increased rapidly in the context of price strategy, and then showed different downward trends according to different price strategies. The stronger the price strategy, the greater the market share of these two regions. If the price strategy is not adopted, that is, under the situation of no price discounting, although the attraction of Lijiang and Diqing is high, the tourism cost is also high, their market share will gradually decline over time, so that although the market share shows a trend of first rising and then falling, the final market share will not be improved compared with the initial one.

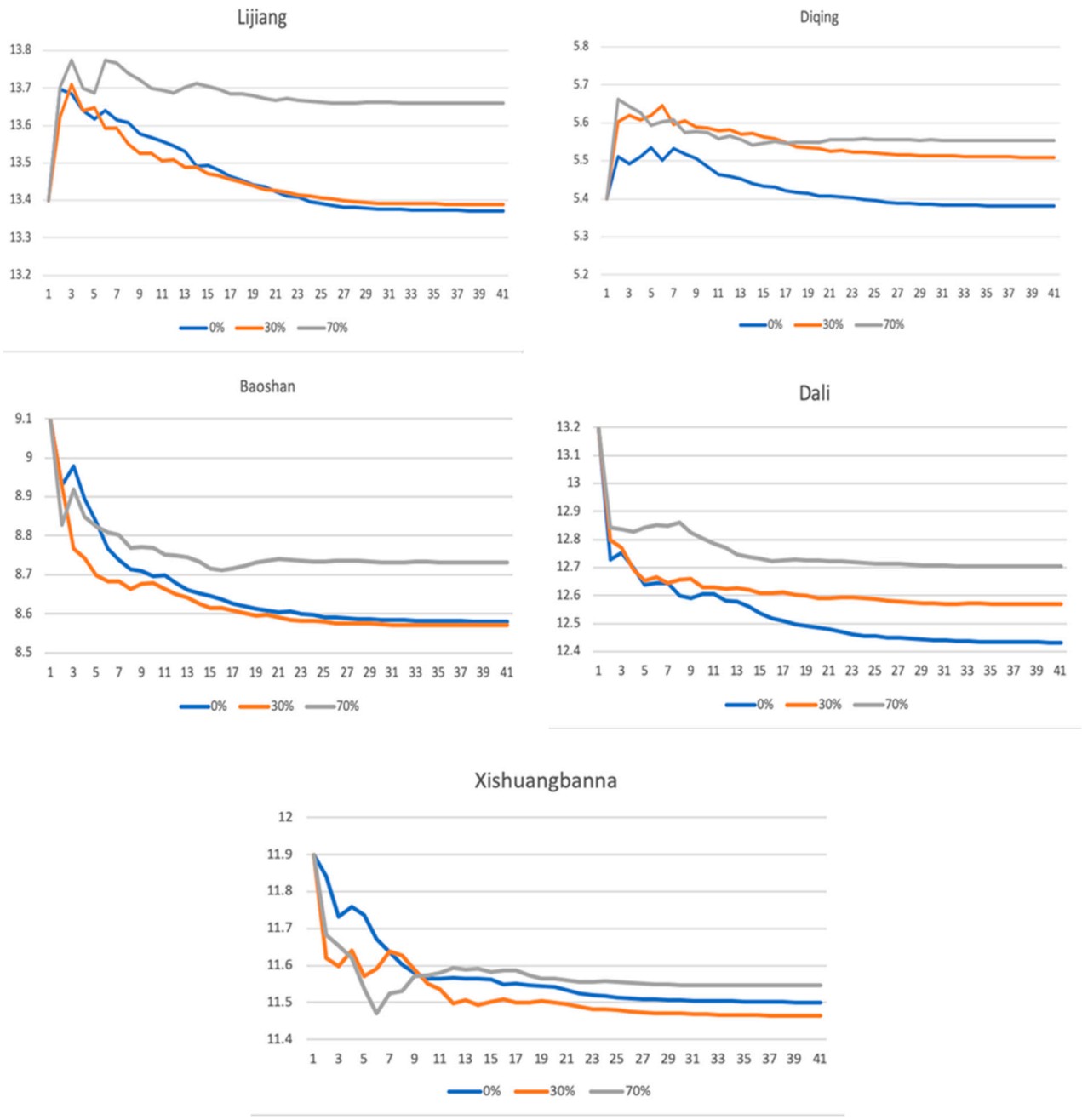

**Figure 8.** Market share of each region under different price strategies.

After the epidemic, although the market share of Dali, Xishuangbanna, and Diqing showed a downward trend, the recovery effect of price strategy on market shares was still shown, that is, the stronger the price strategy, the less the decline of market share. Specifically, the market share with a 70% price strategy is significantly higher than that of a 30% price strategy.

### 5.3. Comprehensive Strategy Analysis

When price strategy and information strategy are applied simultaneously, there are four possible strategy combinations. Figures 9–13 shows the tourist market share changes in four strategy combinations, in which p1 and p2 represent low and high strength information strategies, respectively, and 30% and 70% represent 30% and 70% price discounts in price strategy, respectively.

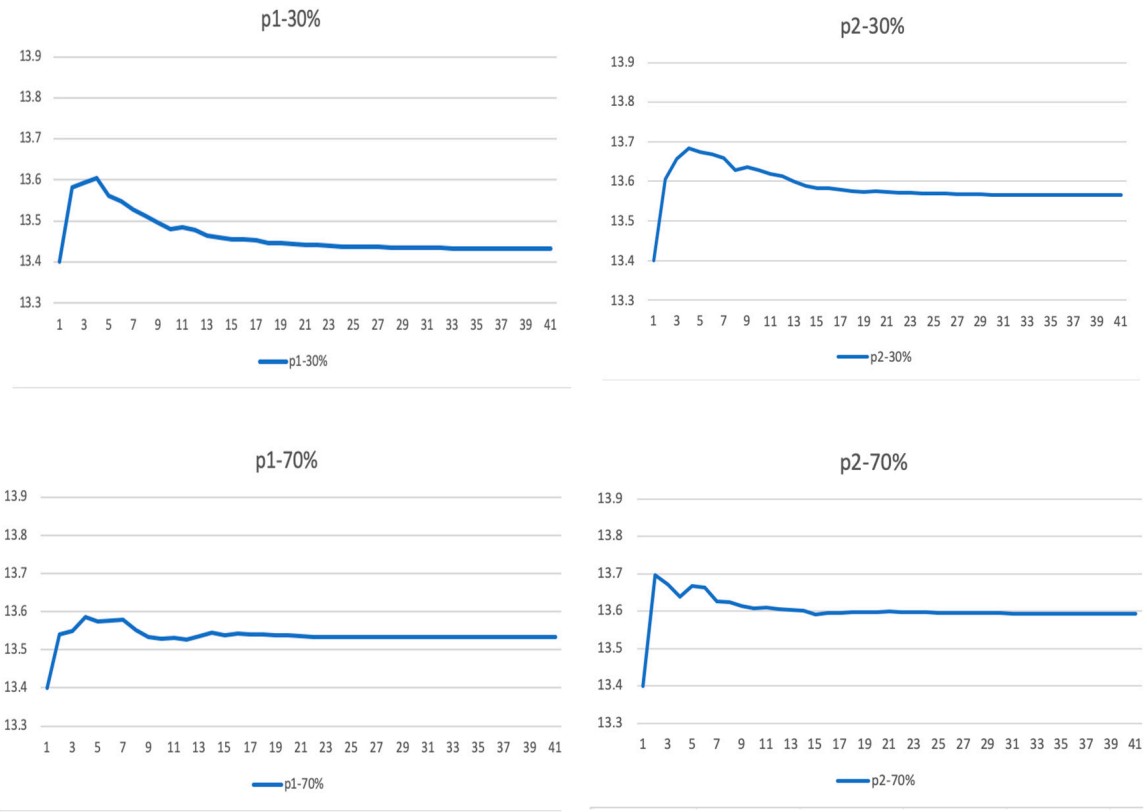

**Figure 9.** Market share of Lijiang under different price and information strategies.

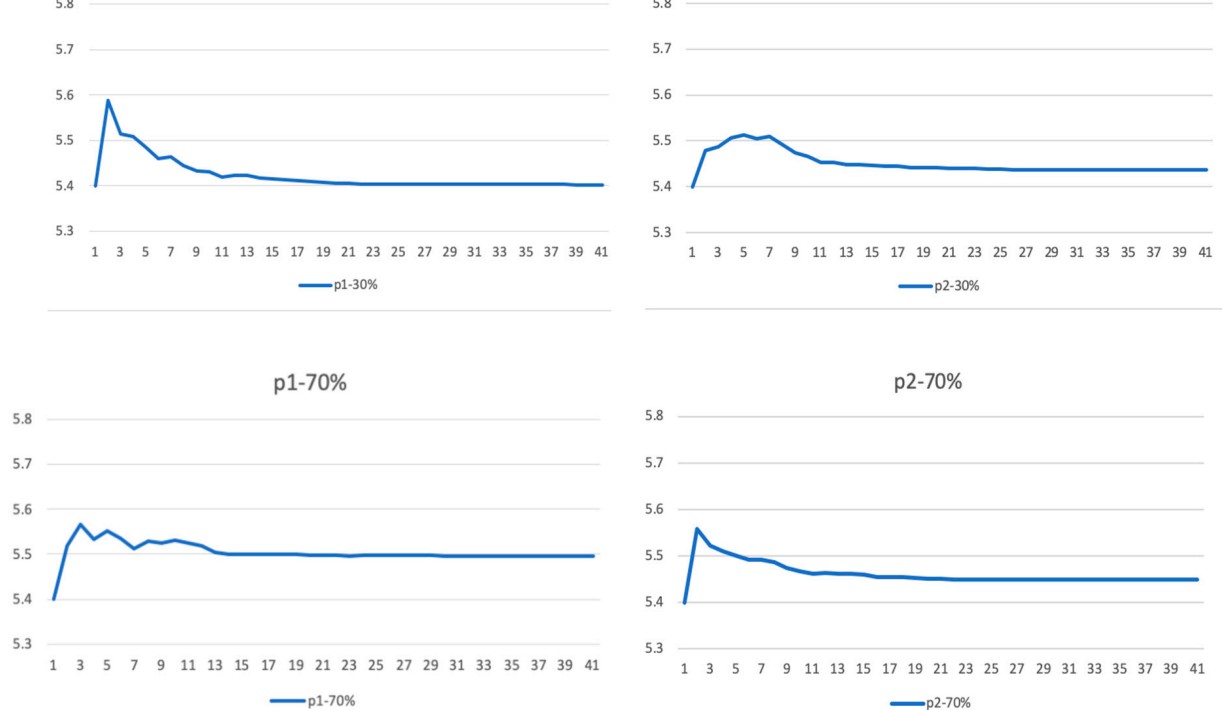

**Figure 10.** Market share of Diqing under different price and information strategies.

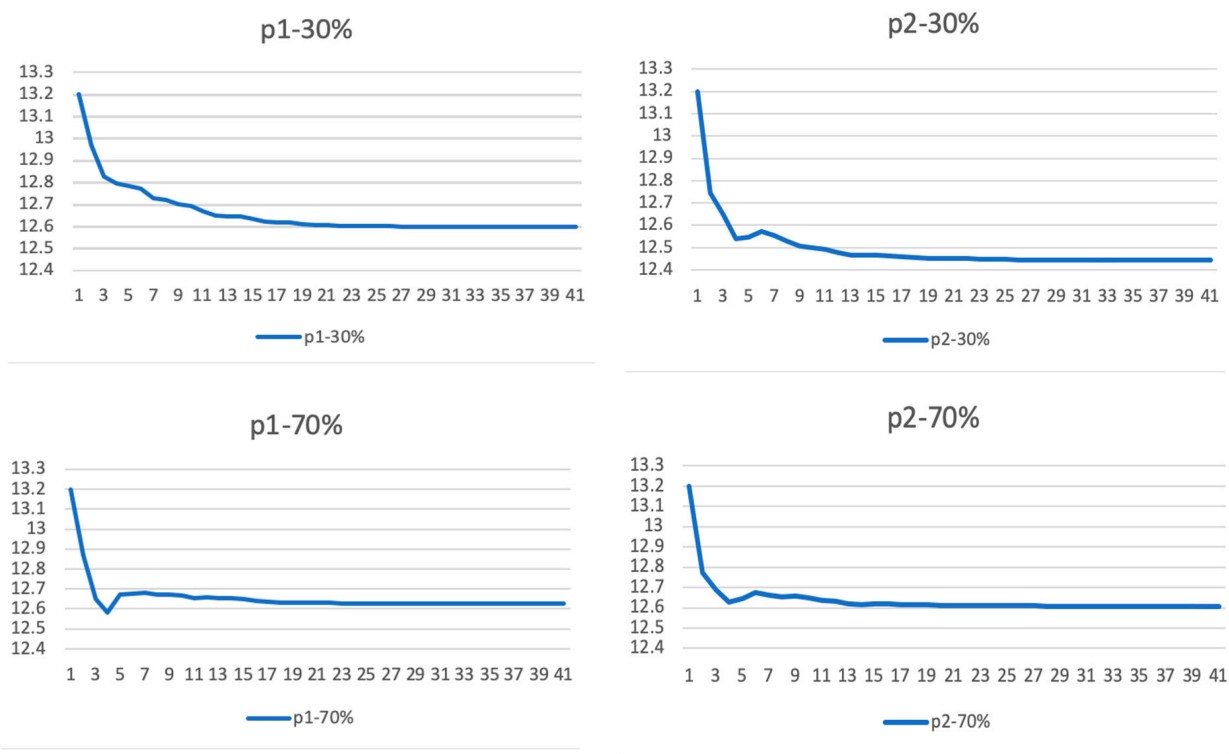

**Figure 11.** Market share of Dali under different price and information strategies.

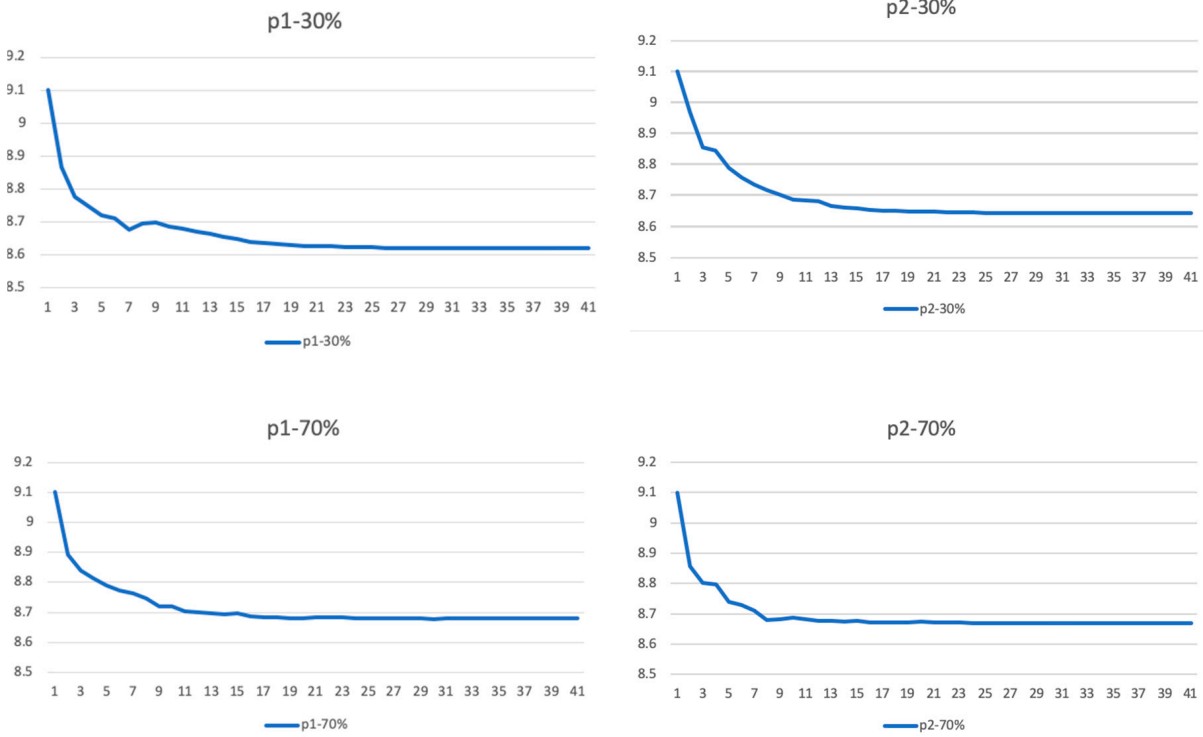

**Figure 12.** Market share of Baoshan under different price and information strategies.

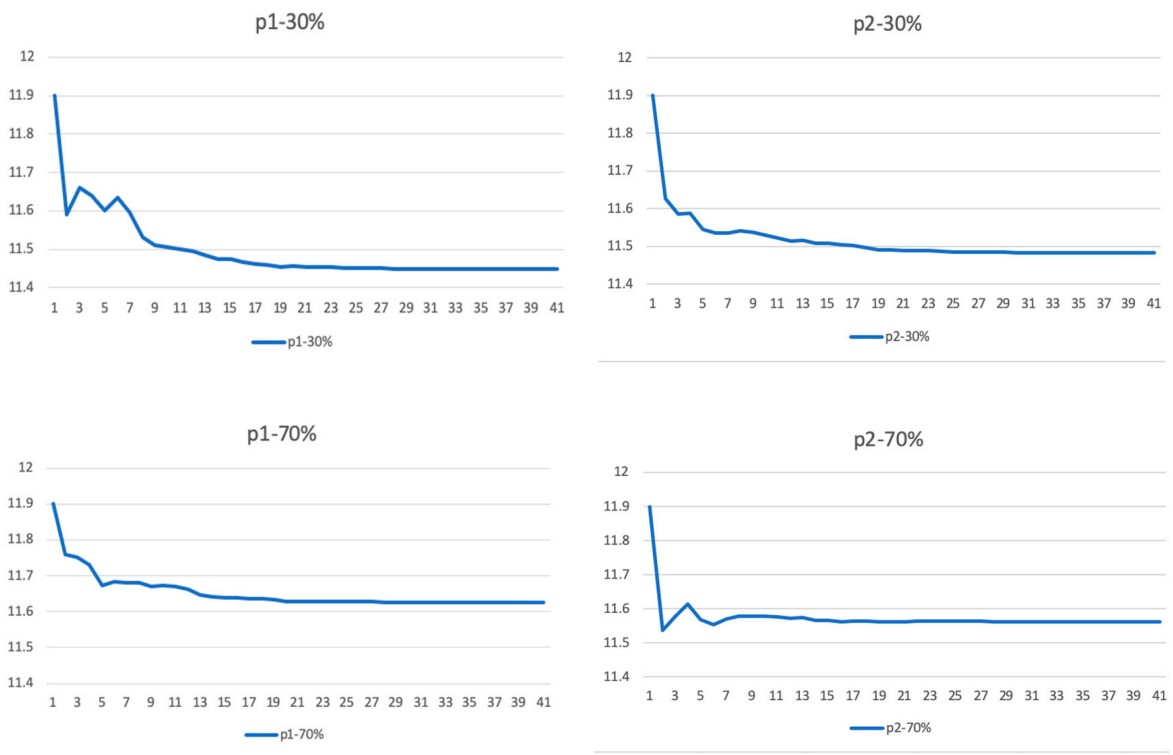

**Figure 13.** Market share of Xishuangbanna under different price and information strategies. Notes: p0 = no information strategy, p1 = low information strategy, p2 = high information strategy, 0% = no price discounting, 30% = 30% price discounting, 70% = 70% price discounting.

Figure 9 shows that in Lijiang, the combination of information strategy and price strategy with the best market recovery effect is p2 and 70%, and the worst combination is p1 and 30%. In Dali, the best combination is p1 and 70% and the worst is p2 and 30% (see Figure 11). In Diqing (see Figure 10), Baoshan (see Figure 12), and Xishuangbanna (see Figure 13), the best combination is p1 and 70%, and the worst combination is p1 and 30%. Overall, for the optimal combination strategy, the information strategy differs and the price strategy is 70%. Therefore, the higher the degree of adopting information strategy is not necessarily better, but the higher the intensity of the adopting price strategy is. The market share in Lijiang and Diqing regions first increases and then decreases, while the market share in Dali, Xishuangbanna, and Diqing regions shows a downward trend. Further, the interaction effects of price strategy and information strategy are inconsistent in different destinations. Specifically, the market recovery effect of price strategy is negatively moderated by information strategy in Dali, while it is positively moderated in other destinations. In Lijiang, Diqin, Baoshan, and Xishuangbanna, in the context of a p1 information strategy, the positive relationships between price strategy and market share become stronger than in the p2 information strategy. By contrast, in the Dali region, the positive relationship between price strategy and market share become weaker under a p2 information strategy.

## 6. Discussion

This paper studied the decision-making process of tourism destination based on an agent-based simulation method. Boavida et al. [40] consider the tourist's motivation, rational behavior rules, emotion and satisfaction, as well as the influence of social networks, and Alvarez and Brida [24] consider tourists' preference for products and services, crowding types, and individual inertia levels. Unlike these studies, this study adopted a multi-factor influence model based on SOR theory, and divided the influencing factors on destination decision after COVID-19 into tourists' internal factors and external factors, considering

tourists' preference, risk perception, time budget, expense budget, and the destination attractiveness, popularity, cost level, and the best days to visit.

The world has experienced several major epidemics in the last 40 years. However, the impact of the COVID-19 epidemic and recovery from it are unprecedented. International, regional, and local travel restrictions immediately affected tourism systems [52]. This paper mainly studied the dynamic recovery process of destinations under different price and information strategies. Certain destinations seem to be more resilient than others both in terms of their ability to adapt to change and their speed of recovery from a crisis [25].

This study found that after the epidemic, for destinations with high attractiveness, such as Lijiang and Diqing, and that have strong competitiveness in the market, their market share will recover in a short time, and then gradually decline. For destinations with relatively low attractiveness, such as Dali, Xishuangbanna, and Diqing, their overall market is greatly affected by the epidemic and shows a downward trend.

Government and scenic managers adopt a series of information strategies to alleviate tourists' risk perception that is affected by the outbreak of COVID-19. Residents negative emotions are alleviated and they adopt a positive outlook if the government performs well in controlling an epidemic [53,54]. However, intense information strategies may lead populations to overreact to an epidemic [52]. A conclusion of this paper is that information strategy has a recovery effect on the final stable market share of all regions, but the market recovery effect of the information strategy is mixed. Due to the low attraction and popularity of Baoshan, an information strategy needs be adopted, while Dali and Xishuangbanna have relatively low attraction and high popularity, so it is recommended that they adopt a low-strength or no information strategy. Xishuangbanna is located at the border, and government intervention will increase users' perception of risk, so it is more appropriate not to adopt an information strategy. Lijiang, with high attractiveness and popularity, should not adopt an information strategy, while Diqing with low popularity but high attractiveness, could only adopt a low-intensity information strategy to improve the market. In addition, based on low popularity, the impact of information strategy on Baoshan's and Diqing's market share is limited.

The epidemic has severely damaged the tourism industry and preferential measures for travelling are needed to stimulate tourist demand [55]. The conclusion of this paper for the price strategy, after the epidemic, is that only the implementation of price strategy has a great positive effect on the recovery of market share in each region. It is consistent with Andreea Orîndaru's [56] finding that tourists may want to reduce travel spending after the pandemic by re-orientating towards more affordable destinations or towards lower transportation costs. Furthermore, the greater the strength of the price strategy, the better the recovery effect of market share. Therefore, price strategy can significantly influence recovery of the tourism market.

With respect to combination of strategies, the market recovery effect of price strategy is negatively moderated by information strategy in Dali, while it is positively moderated in other destinations. Therefore, only adopting a high price strategy or adopting a low information strategy is the most effective in Dali, while in other regions, a comprehensive strategy is more beneficial for market recovery.

## 7. Conclusions

Scientific and effective crisis management is of strategic significance for reducing the economic and social losses caused by tourism crisis events and promoting the sustainable and healthy development of the tourism industry. However, previous studies have paid insufficient attention to the characteristics of tourist heterogeneity and the dynamics of the tourism system, and have lacked verification of the effectiveness and applicable conditions of market recovery strategies, and it is been difficult to predict the effect of the strategy and its dynamic evolution trend. Therefore, we chose Yunnan, China, as an example, and used the agent modeling method to construct a dynamic model with heterogeneous tourists and interactions between the agents. For Yunnan, as an example, the simulation results

show, firstly, that compared with less attractive areas, the more attractive regions have higher market competitiveness, and their market share will increase in a short period of time. Second, when a price strategy or information strategy is adopted alone, the recovery effect of the price strategy is better, and the direct linear effect is obvious, while different information strategies will affect the market share, but the improvement effect does not differ greatly between them. Third, when a comprehensive recovery strategy is adopted, the market recovery effect of the price strategy is negatively regulated by the information strategy in Dali, while it is positively regulated in other destinations.

The results of this study have theoretical and practical significance. In terms of implications for theory, it provides new research methods for the tourism market recovery strategy and supplements the research on tourism crisis management. Previous studies on tourism crisis management mainly assumed that tourism activities are static, linear, and homogeneous, and that it is difficult to evaluate the applicability and implementation effects of strategies [57]. Through the application of ABM, our research has shown that, it is feasible to predict the results of the recovery strategy from the perspective of system, heterogeneity and dynamics. This successfully bridges the gap highlighted by previous research [15].

In terms of implications for practice, managers can find a suitable combination of strategies to restore the market based on tourists' responses to different recovery strategies. Previous studies have put forward many recovery strategies or suggestions after earthquakes. However, these strategies are difficult to implement because they do not provide quantitative and practical guidance. For example, Liu and Li [57] proved that the strategy combination of safety, price, experience, and emotion is effective overall, but they did not give clear suggestions on how to quantify these strategies. The study found that price strategy can enable significant recovery and increase the destination market share. Therefore, if managers want to realize a market rebound as soon as possible, they can adopt a high-strength price strategy. For different destinations, the recovery effect of the information strategy is positive, negative, or none. Therefore, the manager should fully consider the market recovery effect and operating cost of the strategy and adopt different combination strategies in different tourism destinations to ensure optimal market recovery.

The limitations of this research primarily relate to the following issues: first, this study examines certain factors affecting the decision-making of tourists' destinations after the epidemic, but whether there are other factors that affect the decision-making of tourists' destinations after the epidemic, and whether there are other interactions between these factors, remains to be further explored; secondly, the data source is relatively limited. Due to the complex factors involved in the tourism system, it is difficult to accurately set the variables and parameters of the model with limited data. It only provides quantitative information on trends and cannot achieve accurate prediction. This suggests a direction for future research for continued improvement.

**Author Contributions:** Conceptualization, Y.L. (Yumei Luo) and Q.Y.; methodology, Y.L. (Yumei Luo) and Y.L. (Yuwei Li); software, Y.L. (Yuwei Li); data curation, G.W.; writing—original draft preparation, G.W. and Y.L. (Yuwei Li); writing—review and editing, Y.L. (Yumei Luo), G.W. and Y.L. (Yuwei Li); funding acquisition, Y.L. (Yumei Luo) and Q.Y. All authors have read and agreed to the published version of the manuscript.

**Funding:** This work was supported by the Kunming E-commerce and Internet Finance R&D Center #1 under Grant KEIRDC, 2020; Prominent Educator Program #2 under Grant Yunnan [2018] 11; Yunnan Province Young Academic and Technical Leader candidate Program #3 under Grant 2018HB027; and the 2020 scientific research project of Yunnan philosophy and social science innovation team; 2020 Yunnan Philosophy and Social Science Innovation team scientific research project #5.

**Institutional Review Board Statement:** Not applicable.

**Informed Consent Statement:** Not applicable.

**Data Availability Statement:** Publicly available datasets were analyzed in this study. This data can be found here: www.ctrip.com (accessed on 21 October 2021) and http://stats.yn.gov.cn/tjsj/tjnj/ (accessed on 21 October 2021).

**Conflicts of Interest:** The authors declare no conflict of interest.

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
