# Peer review of "Agent-Based Modeling and Simulation of Tourism Market Recovery Strategy after COVID-19 in Yunnan, China"

_sustainability, doi:10.3390/su132111750_

Round 1

Reviewer 1 Report

This article presents a systems modeling take on the recovery strategy of tourism market amid the Covid-19 pandemic in China. It is a timely issue at which the Chinese tourism industry needs supports from systems modelers to understand projections induced by potential post-pandemic strategies. At a first look, this article presents an excellent execution of systems modeling and simulation using Agent-Based Modeling (ABM). Besides, this article attempts to include as many as possible results gathered during the conduct of the research. However, I am afraid the current form of the manuscript inadequately explains the build-up of critical thinking that lead to the developments of the research and the ABM model. It results in a lack of clarity and linearity, and an unclear engagement of this study to relevant tourism literature. Therefore, the authors must address some critical issues before this article can be reconsidered for publication.

  1. Language concern. This article requires an extensive English proofreading/editing by a native speaker. Some sentences are confusing, unnecessarily long, and sometimes incomplete (e.g. Line 314: "As shown in ...").
  2. Introduction section.
    • This article attempts to deal with a practical problem in the Chinese tourism industry from a systems modeling (ABM) perspective. It is apparent from the mention of China-related data and information throughout the article, including the majority of references focusing on China-related topics/cases. However, the authors are indecisive on whether to address a practical problem in the country's tourism industry or a theoretical problem in the tourism literature. ABM has been widely used in tourism, marketing, and related topics; thus addressing the issue as a theoretical problem limits the contribution of this research.
    • In this Introduction section, the authors must include more data and information on China-related tourism industry, how the industry deals with the pandemic, and potential interventions against the impact of the pandemic. The authors should cite relevant literature that supports those arguments/information. Please refer to the last point of this review for your consideration. This section should end with the Research Gap(s) and Research Aim(s) implying the necessity of this research from the perspective of the Chinese tourism industry.
    • Research Questions should also be provided in the end of this section to indicate specific questions arising from the gap(s) and aim(s). These questions would be the focuses to address by this research.
  3. Literature Review section.
    • The current literature review inadequately builds the arguments for a Research Framework. Please add another subsection for the Research Framework at the end of this section, which would compare-and-contrast potential issues to be included later in the systems model and scenario building.
    • Lines 124-133. This paragraph lacks references to support each argument.
    • In general, this Literature Review section includes extremely limited number of literature being reviewed. Please refer to the last point of this review for your consideration.
  4. Methodology section.
    • I am surprised that this article does not include any Methodology section. Readers would not understand how the research was conducted, or whether it was properly conducted or not.
    • The authors must include the section, which includes (at least) "Research Design" and "Data Collection" subsections.
      • The Research Design subsection should include the stages of this research from the research framework to the interpretation of results. The design should include the objective(s) of, technique(s) used in, and expected outcome from each research stage.
      • The Data Collection section should include explanations on data types being used, data sources, and data collection methods being used.
  5. Model Building section.
    • For the section currently titled "Developing an Agent-Based Model", please add the phrase "Model Building:" at the beginning of the title.
    • Lines 257-290. The current explanations on decision-making logic are difficult to follow by readers. Please use equation-style presentation for the decision logic for a clearer explanation.
  6. Model Testing section.
    • For the section currently titled "Simulation system parameterization, calibration and validation", please add the phrase "Model Testing:" at the beginning of the title.
    • Table 1. Please use an equation-style presentation to explain each calculation method. If spaces do not allow, please add them in the text and add references to the equations in the table.
    • Table 1. It is difficult to distinguish between information pairs (rows). Please add lines for a clearer presentation (similar to that of Table 2).
  7. Scenario Testing section.
    • For the section currently titled "Analysis of simulation results", please add the phrase "Scenario Testing:" at the beginning of the title.
    • There is a limited explanation of the settings and assumptions of each scenario testing. Please add the settings and assumptions in a table for each scenario testing.
    • Figures 5-10. Please use larger figures for those graphs, and increase the font size to improve their readability.
    • In general, this section lacks more explanations on the results of each testing settings-assumptions. The explanations should refer to the settings and assumptions provided above.
  8. Discussion section.
    • I am again surprised that this article does not include a Discussion section. This section should be the place where the authors argue the contribution of this China-focused research to the body of knowledge of relevant tourism themes.
    • The authors must add the section by conducting a compare-and-contrast of each finding of this research to the results of relevant tourism research in the scholarly literature.
  9. Conclusion section.
    • Please merge "7. Contribution" and "8. Limitations" into a single Conclusion section.
    • Please separate mentions of key findings and managerial implications.
      • The first paragraph should focus on the summary of the research and its key findings.
      • The second paragraph should focus on managerial and policy implications arising from those key findings.
      • The third paragraph should focus on the limitations and insights for further research.
  10. Abstract section.
    • Please rewrite the Abstract to present the article structure of this China-focused tourism research.
    • The abstract has not included the Research Gap(s) being addressed (in the beginning) and the original contribution as well as managerial/policy implications (in the end).
  11. References section.
    • Many references listed in this section do not provide complete citation data. Please make sure that they are properly cited.
    • There are duplicated references on the list. Please only list one source for one referred literature.
  12. In-text references.
    • Lines 26-28. Please cite the source for UNWTO data referred to in the sentence.
    • Line 298. Please cite which page(s) on ctrip.com and the "Yunnan statistical yearbook" at the end of this sentence.
    • There are duplicated in-text citations in the same locations (e.g., Lines 81, 85, etc.). Please make sure that every location for in-text references does not include duplicated citations for a single source.
  13. Engagement with relevant literature. I am again surprised that this research does not engage with up-to-date tourism literature and Covid-19-related topics. Please search thoroughly for these topics/issues and engage with relevant literature in the Introduction, Literature Review, and Discussion sections.
    • This list is not exhaustive. Considering the timely issue of post-pandemic tourism recovery, the authors must ensure that this article properly engages with relevant literature.
    • Forecasting tourism recovery amid COVID-19 (https://doi.org/10.1016/j.annals.2021.103149).
    • The good, the bad and the ugly on COVID-19 tourism recovery (https://doi.org/10.1016/j.annals.2020.103117).
    • Sustainable and resilient strategies for touristic cities against COVID-19: An agent-based approach (https://doi.org/10.1016/j.ssci.2021.105399).
    • Small Island Developing States (SIDS) COVID-19 post-pandemic tourism recovery: A system dynamics approach (https://doi.org/10.1080/13683500.2021.1924636).
    • Managing relationships in the Tourism Supply Chain to overcome epidemic outbreaks: The case of COVID-19 and the hospitality industry in Spain (https://doi.org/10.1016/j.ijhm.2020.102733).
    • etc.

Reviewer 2 Report

The paper analyzed the recovery of Tourism under the outbreak and spread of COVID-19 considering an agent-based simulation method to simulate the decision-making process of tourists' destinations selection and the dynamic recovery process of the destinations under different price strategies and information strategies. The authors present a study was revealed the decision-making process of tourists' destinations from a dynamic perspective intending to show the dynamic recovery process of the destination for different levels of recovery strategies, such as information strategies and price strategies. This topic is broad of international interest and expectations of this topic are high in different countries.

Article and topic are certainly interesting and within the scope of the journal. I have, however, some substantial concerns about the paper as it is presented - to name the most important issues:

Constructive feedback:

References are in APA standards and not in those accepted by the journal and the MDPI publisher. Authors must list each reference as it appears in the text.

The section named "Literature review" is very poor. There are many articles to show how is tourism crisis management currently.

The Introduction is a compilation of diverse information on the topic that is loosely connected with each other. Therefore, the background and rationale of this study did not become clear.

The research design, questions, hypotheses, and methods of research are not clearly stated. Perhaps a methodological scheme of the research could be designed. I recommend this idea to the authors.

Results and discussions sections are not included and separated.  Results are very briefly presented. This is mostly limited to explanations in the text, while only very few results are presented in the several figures. The findings are poorly connected so that no greater picture is developed, but numerous small details are shown. Discussion of results is very weak in the article. Few references have been analyzed and quoted.

I recommend the re-submission of this paper.

Reviewer 3 Report

The theme analyzed with methodological rigor. The work structure is good and well articulated.

Literature Review. Insert further bibliographical references

The results are explained in a clear and detailed manner.

Overall the work is original.

It is required to implement the conclusions and make them more incisive.

Reviewer 4 Report

The paper is written well and the analysis is robust. The chosen topic is appropriate and timely. The overall quality is good with evidence of engagement and background research. Few weaknesses which need to be considered. Firstly, the paper needs more attention in terms of English, referencing, etc. There are few spelling and grammar issues, some sub-headings begin with lower cases and the referencing is inconsistent (see for example, the reference list where some journal titles are in capitals and others are not). Some sources are repeated twice in the same sentence (e.g. Fen et al. 2020) which is not needed. 

Secondly, the literature review is rather weak at the moment. Most of it is about crisis management and previous crises but there is a limited discussion on crisis management frameworks. Further, the section does not provide an up-to date review of papers published on tourism and COVID-19 from a crisis management perspective. Such studies do exist in the wider literature on tourism, management and COVID-19.

Round 2

Reviewer 1 Report

Most of the substantive changes have been addressed. I would like to recommend this article for publication given the following minor revisions.

  1. Title. Please remove the words "Research on" from the title since they are unnecessary. Instead, add a phrase "in Yunnan, China" at the end of the title to indicate the location of observed cases of this particular research.
  2. Case Description (3.1). Please add a Figure to indicate the location of Yunnan in China, the location of Kunming in Yunnan, and the location of the observed cases (Lijiang, Dali, Baoshan, Xishuangbanna, Diqing). The authors may consider using a couple of maps in one figure to indicate the relationship between administrative levels.
  3. Discussion (section). The addition of this section is much appreciated. However, the added section does only contain one reference to existing literature. Please enrich the discussion by doing more compare-and-contrast processes of the results of this research to relevant research in "tourism recovery/crisis management" and "ABM" topics. As the place where the authors argue the contributions of their key findings to relevant body of knowledge, this section is typically longer than the Conclusion section.
  4. References (throughout the text). It is still unclear how strong this research connects to existing scholarly discourses in the Sustainability journal. Please clarify this by citing previous works published in this journal that are relevant to the content of this article.

I appreciate all efforts made by the authors to address the concerns of the reviewers. I would like to say a piece of good luck with the publication, including the continuity of research on tourism topics using systems modeling methods.

Reviewer 2 Report

The authors have improved the manuscript. However, quoted references continued very poorly. I recommend that the authors cite some of these articles to enrich their section for introduction and discussion of results. It is important that the authors reveal the global impact of COVID-19 on biodiversity, tourism, and lagoon and coastal management because COVID-19 has a relevant impact on tourist activities. 
